# Effects of Simulated In Vitro Digestion on the Structural Characteristics, Inhibitory Activity on α-Glucosidase, and Fermentation Behaviours of a Polysaccharide from *Anemarrhena asphodeloides* Bunge

**DOI:** 10.3390/nu15081965

**Published:** 2023-04-19

**Authors:** Juncheng Chen, Meijuan Lan, Xia Zhang, Wenjuan Jiao, Zhiyi Chen, Lin Li, Bing Li

**Affiliations:** 1International School of Public Health and One Health, Hainan Medical University, Haikou 571199, China; chenjuncheng2009@163.com; 2School of Food Science and Engineering, Guangdong Province Key Laboratory for Green Processing of Natural Products and Product Safety, Engineering Research Center of Starch and Plant Protein Deep Processing, Ministry of Education, South China University of Technology, Guangzhou 510640, China; 3Sericultural & Agri-Food Research Institute, Guangdong Academy of Agricultural Sciences Key Laboratory of Functional Foods, Ministry of Agriculture and Rural Affairs, Guangdong Key Laboratory of Agricultural Products Processing, Guangzhou 510610, China; 4Food Chemistry and Technology, College of Oceanology and Food Science, Quanzhou Normal University, Quanzhou 362000, China

**Keywords:** polysaccharide, *Anemarrhena asphodeloides* Bunge, salivary–gastrointestinal digestion, gut microbiota

## Abstract

The purpose of this study is to investigate the effects of the simulated saliva–gastrointestinal digestion of AABP-2B on its structural features, inhibitory α-glucosidase activity, and human gut microbiota. The salivary–gastrointestinal digestion results show that there is no significant change in the molecular weight of AABP-2B, and no free monosaccharides are released. This indicates that, under a simulated digestive condition, AABP-2B is not degraded and can be further utilized by gut microbiota. AABP-2B still possessed good inhibitory activity on α-glucosidase after salivary–gastrointestinal digestion, which may be attributed to the largely unchanged structural characteristics of AABP-2B after simulated digestion. Furthermore, in vitro fecal fermentation with AABP-2B after salivary–gastrointestinal digestion showed that AABP-2B modulated the gut microbiota structure and increased the relative proportions of *Prevotella*, *Faecalibacterium*, and *Megasphaera*. AABP-2B can also modify the intestinal flora composition by inhibiting pathogen growth. Moreover, the AABP-2B group resulted in a significant increase in short-chain fatty acid (SCFAs) content during fermentation. These findings demonstrate that AABP-2B can be used as a prebiotic or functional food to promote gut health.

## 1. Introduction

Trillions of microbial cells are distributed in the human gastrointestinal tract, forming a large microbial ecosystem that plays an important role in human health, such as regulating intestinal endocrine function, preventing the excessive growth of pathogens, and promoting the maturation of host immune responses [1,2,3]. Conversely, an imbalance in the intestinal flora could induce several diseases, such as intestinal barrier disorders, diabetes, and liver disease [4]. In recent decades, the research exploring the link between gut microbiota and human health has rapidly progressed, generating significant interest from both the scientific community and general public. Thus, the gut microbiota has emerged as a promising target for improving human health and managing diseases. In addition to supplying the essential nutrients for the human body, diet serves as a critical factor in maintaining the survival of trillions of gut microbiota. Consequently, dietary intake plays a key role in influencing the composition and function of the gut microbiota [5]. Dietary polysaccharides have been reported to resist to be digested in the upper gastrointestinal (GI) tract and sequently enter the large intestine without causing dramatic structural changes. Consequently, the flora in the large intestine may utilize dietary polysaccharides to produce beneficial metabolites and regulate pH. Previous studies have reported that polysaccharides from okra [6] and loquat leaves [7] modulate the composition of the intestinal flora, particularly by promoting certain beneficial intestinal flora, such as *Lactobacillus* and *Prevotella*. Dietary polysaccharide intervention tends to transform the intestinal microbiota composition, and, concomitantly, the metabolites produced by this process contribute to the health of the host.

*Anemarrhena asphodeloides* Bunge, also called Zhimu, has long been regarded as a functional food or Chinese herb in China in light of its beneficial biological properties, such as antioxidant, hypoglycaemic, and anti-inflammatory entities. Polysaccharides are important active ingredients of *Anemarrhena asphodeloides* Bunge and exhibit neuroprotective and hypoglycaemic activities [8]. Additionally, four types of hypoglycaemic glycans have been obtained from *A. asphodeloides* [9]. Our previous study isolated a polysaccharide (AABP-2B, 5800 Da) from *A. asphodeloides*, which mainly included (4,6)-β-d-Glcp-(1,3,6)-d-Manp-(1,4)-β-d-Manp-(1) glycosidic linkages. AABP-2B not only possessed favourable α-glucosidase-inhibitory activity, but also clearly improved insulin resistance [10]. Postprandial hyperglycaemia is a common symptom of diabetes mellitus. Floris et al. [11] showed that inhibiting the activity of α-glucosidase could delay the conversion of dietary carbohydrates into glucose entering the bloodstream, thereby reducing postprandial hyperglycaemia. In addition, an increasing number of studies have shown that the aetiology or progression of T2DM is related to intestinal flora disorders. Unbalanced microbial communities can ultimately destroy the intestinal barrier function, resulting in insulin resistance and obesity [12,13]. Based on the structural characteristics of AABP-2B, it is thought that AABP-2B (a non-starch polysaccharide) may not be completely degraded by gastrointestinal digestion and enter the colon for fermentation by the intestinal flora. It has yet to be determined, however, whether AABP-2B still has an inhibitory effect on α-glucosidase [10] after encountering carbohydrate-hydrolysing enzymes and whether fermentation of AABP-2B by intestinal flora can benefit the host.

Hence, the purpose of this study is to investigate the effects of the simulated saliva–gastrointestinal digestion of AABP-2B on the structural features and inhibitory α-glucosidase activity. Furthermore, the effects of indigested AABP-2B on the intestinal flora and the production of SCFAs were studied using in vitro fermentation models. The results of this research will be helpful for better understanding the potential digestion and fermentation mechanism behind AABP-2B, providing a theoretical basis for its development and application in functional food.

## 2. Materials and Methods

### 2.1. Materials and Chemicals

*Anemarrhena asphodeloides* Bunge was provided by Chengji Pharmaceutical Co. (Guizhou, China). Pepsin (3000 U/g), pancreatin, α-glucosidase, α-amylase, trypsin, and SCFA standards were purchased by Sigma-Aldrich Chemical Co. (St. Louis, MO, USA). Bile salts and inulin were obtained from Yuanye Biotechnology Co., Ltd. (Shanghai, China). The TIANamp Stool DNA Kit was provided by Tiangen Biotech Co. (Beijing, China). All other reagents and solvents used in this study were of analytical grade.

### 2.2. Preparation of AABP-2B

AABP-2B was extracted as in our previously described method [10]. *A. asphodeloides* rhizome powder was extracted 3 times for 4 h with distilled water (1:35, *w*/*v*) at 90 °C, filtered, concentrated, and precipitated with absolute alcohol (80% alcohol) to obtain crude polysaccharide (AABP). After deproteinising AABP, it was purified by DEAE-cellulose chromatography, dialysed, and freeze-dried to obtain AABP-2B.

### 2.3. Oral Digestion Simulation

The simulation of oral digestion was performed according to the published literature [14], with slight modifications. Briefly, three healthy volunteers without chronic diseases or antibiotic use within the past three months were selected. The volunteers did not eat or drink before saliva collection. The volunteers then rinsed their mouths three times with deionised water and discarded the first 30 s of saliva. The saliva was pooled and centrifuged (12,000× *g*) and the supernatant collected. The α-amylase activity in the saliva supernatant was determined. The AABP-2B solution (20 mL of 15.0 mg/mL) was blended with the saliva (20 mL). The samples were then placed in a shaker in a water bath at 37 °C. During digestion, 2.0 mL of digestive fluid was removed at 0 and 0.5 h, respectively, and boiled for 10 min to destroy enzyme activity.

### 2.4. Simulation of Gastric Digestion

Gastric digestion was simulated, as described previously, with slight modifications [1]. The simulated gastric electrolyte solution was prepared with 775.0 mg NaCl, 37.5 mg CaCl_2_·2H_2_O, 275.0 mg KCl, and 150.0 mg NaHCO_3_ into 250 mL deionized water. Then, 37.5 mg of gastric lipase, 35.4 mg of pepsin, and 1.5 mL of CH_3_COONa solution were added to the prepared 250 mL gastric electrolyte solution. After stirring, the final pH of the produced gastric juice was modified to 3.0 using a 0.1 M HCl solution. The saliva-digested AABP-2B solution was blended with an equal volume of simulated gastric juice, and the samples were then placed in a shaker in a water bath at 37 °C. During digestion, 2.0 mL of the digestive fluid was sampled out at 0, 2, and 4 h and boiled for 10 min to destroy the enzyme activity. The remaining samples were digested with the small-intestinal juice.

### 2.5. Simulation of Small-Intestinal Digestion

The intestinal electrolyte solution’s preparation and digestion were based on a previously reported procedure with some modifications [15]. The simulated small-intestinal electrolyte solution was prepared with 1.35 g NaCl, 162.5 mg KCl, and 82.5 mg CaCl_2_·2H_2_O into 250 mL deionized water, and the pH of the solution was adjusted to 7.0 using 0.1 M NaOH. Then, 20.0 g of pancreatic enzyme solution (7%, *w*/*w*), 40.0 g of bile salt solution (4%, *w*/*w*), and 65.0 mg trypsin were added to the prepared 20.0 g small-intestinal electrolyte solution. After stirring, the final pH of the simulated small-intestinal juice was adjusted to 7.0 using a 0.1 M NaOH solution. First, the pH of the mixture was adjusted to pH 7.0 by adding 0.1 M NaOH after gastric digestion. Equal volumes of artificial small-intestinal fluid were added. The mixture was placed in a shaker in a water bath at 37 °C. During digestion, 2.0 mL of the digestive fluid was sampled out at 0, 2, and 4 h and boiled for 10 min to destroy the enzyme activity. Subsequently, the inactivated samples were analysed for molecular weight, monosaccharide constituents, and reducing sugar content.

### 2.6. Analysis of the Monosaccharide Constituents and Molecular Weight of AABP-2B during Digestion

The digested products of AABP-2B were filtered and the filtrate was used to measure the molecular weight by high-performance gel permeation chromatography (HPGPC) [10]. Different molecular-weight dextrans (13.05, 36.8, 64.65, 135.35, 300.6, and 2000 kDa) were used as standards. The chromatographic conditions were as follows: detector: Waters 2414 differential detector. Chromatographic column: Waters TSK G-5000 PW × L TSK Waters (7.8 × 300 mm) and G-3000 PW × L (7.8 × 300 mm) used as two gel chromatography columns in series; mobile phase: 0.02 mol/L KH_2_PO_4_ buffer (pH 6.0). flow rate: 0.6 mL/min. The monosaccharide composition was analysed using PMP-derived HPLC based on our previously published method [16]. A 100 μL volume of 0.6 M NaOH was added to 100 μL of digested solution, mixed well, and then 200 μL of PMP methanol solution (0.5 M) was added and shaken well. After the reaction in a 70 °C water bath for 70 min, the sample was cooled to room temperature, 200 μL 0.3 M HCl solution and 0.5 mL of distilled water were added, followed by 1 mL of dichloromethane, and the extraction was repeated three times. The aqueous phase (upper layer) was passed through a 0.22 μm filter membrane and loaded into a liquid bottle. Finally, the PMP derivatives were analysed by HPLC. The reducing sugar content of AABP-2B during digestion was determined using a reducing sugar content detection kit (BC0235; Solarbio, Beijing, China).

### 2.7. Inhibition of α-Glucosidase Activity

The α-glucosidase inhibitory effects of AABP-2B before and after digestion were determined using a previously reported procedure [10]. A 100 μL aliquot of polysaccharide and 100 μL of α-glucosidase (0.8 U/mL) were mixed in a 96-well microplate and incubated for 30 min (37 °C). The pNPG solution (100 μL, 10 mM) was then added to the 96-well microplate and reacted for 30 min (37 °C), followed by the addition of 100 μL of Na_2_CO_3_ (1 M) solution to terminate the reaction.

### 2.8. Fermentation of AABP-2B In Vitro

Fermentation of AABP-2B was conducted, as previously described, with slight modifications [15]. The basal fermentation nutrient solution contained K_2_HPO_4_ (4 mg), peptone (500 mg), NaCl (2.5 mg), yeast extract (500 mg), CaCl_2_ (2.5 mg), KH_2_PO_4_ (4 mg), MgSO_4_ 7H_2_O (5000 mg), NaHCO_3_ (500 mg), hemin (4 mg), cysteine-HCl (125 mg), Tween 80 (0.5 mL), bile salts (125 mg), 1% resazurin solution (0.25 mL), vitamin K (2.5 μL), and deionised water (250 mL). The pH of the fermentation nutrient was adjusted to 7.0 by adding HCl (0.1 mol/L). Human faeces were obtained from five healthy donors (23–28 years of age) who had not received antibiotics or prebiotics in the past three months. The faeces of five volunteers were collected simultaneously and immediately mixed in equal amounts. Sterile saline was added to make a 10% faecal slurry and then filtered through a nylon gauze under sterile conditions to obtain faecal inoculum. Forty-five millilitres of basal fermentation nutrient solution containing AABP-2B (0.5 g) or inulin (IN, 0.5 g) was blended with 5.0 mL of faecal inoculum and then incubated in an anaerobic incubator at 37 °C. Basal fermentation nutrient solution without AABP-2B was used as the blank group. During the fermentation process, samples were removed at 6, 12, and 24 h. The samples were immediately centrifuged (10,000 rpm, 15 min) at 4 °C, and both the precipitate and supernatant were collected.

### 2.9. SCFAs and pH Determination

The short-chain fatty acid content was measured by gas chromatography-mass spectrometry [14]. First, the fermentation supernatant (1.0 mL) was extracted three times with diethyl ether (1.0 mL each). After centrifugation, the organic phase was collected and concentrated to 1.0 mL for further analysis by gas chromatography-mass spectrometry. The pH of the fermentation supernatants was measured using a pH meter.

### 2.10. Determination of Gut Flora

A TIANamp stool DNA kit was used to extract genomic DNA from all samples, as specified by the manufacturer. Specific primers 338F (5′-ACTCCTACGGGAGGCAGCAG-3′) and 806R (5′-GGACTACHVGGGGTATCTAAT-3′) with barcodes were selected for PCR amplification of the V3–V4 region of bacterial 16S rDNA. The PCR products were first purified using a Monarch DNA Gel Extraction Kit and then sequenced using an Illumina MiSeq system (Shanghai Personal Biotechnology Co., Ltd., Shanghai, China). The raw data were merged and filtered to remove chimeras (UCHIME version 8.1, Tiburon, CA, USA). The raw FASTQ files were disassembled, assembled, and quality-filtered using QIIME software (version 1.17, Flagstaff, AZ, USA) software. The UCHIME algorithm was used to identify and remove chimeric sequences to collect accurate, reliable, and high-quality valid data. Alpha- and beta-diversity values were analysed using QIIME and R software. Cluster analysis was performed using 97% similarity (USEARCH, version 10.0, Tiburon, CA, USA), and operational taxonomic units (OTUs) were screened using a threshold of 0.005% of all sequenced reads.

### 2.11. Statistical Analysis

Each experiment was repeated three times and the experimental results are presented as means ± the standard deviation. One-way ANOVA and Duncan’s test were performed for statistical analysis using the SPSS 22 software (IBM SPSS Software, Chicago, IL, USA). *p* < 0.05 was considered a statistical significance. Origin software (version 8.0, Northampton, MA, USA) was used for the drawings.

## 3. Results and Discussion

### 3.1. An Analysis of the Changes in Molecular Weight (Mw), Monosaccharide Composition, and Reducing Sugar during Saliva–Gastrointestinal Digestion of AABP-2B

In the oral cavity, salivary amylase is the key enzyme responsible for hydrolysing α-1,4 glycosidic bonds in starch or oligosaccharides. Saliva is the primary digestive juice secreted after eating. The amylase activity of human saliva used in this study was measured to be 113 ± 7 D units/mL, which is within the normal range of 18–208 D units/mL [14]. The Mw of AABP-2B after simulated digestion was determined by HPGPC. As shown in Figure 1A, the molecular weight of AABP-2B does not change after 0.5 h of saliva digestion. In addition, free monosaccharides were not detected during digestion (Figure 2B), whereas the reducing sugar content (Figure 3) was not significantly altered, suggesting that AABP-2B cannot be degraded by saliva. 

AABP-2B was digested in vitro in a simulated gastrointestinal environment. In Figure 1B,C, the molecular weight of AABP-2B did not undergo discernible changes during gastric or intestinal digestion. In addition, the changes in monosaccharide compositions of AABP-2B during digestion were investigated. As illustrated in Figure 2, no free monosaccharides were detected during in vitro digestion, indicating that AABP-2B was not digested to release monosaccharides. Moreover, the reducing sugar content of AABP-2B did not change significantly (*p* > 0.05) after the simulation of upper gastrointestinal digestion (Figure 3). These results show that AABP-2B was not digested by the gastric or small-intestinal environment and safely reached the colon. Similarly, the Mw of polysaccharides from Fuzhuan brick tea [17] and tamarind seeds [18] were observed no significant variation (*p* > 0.05) during gastric and small-intestinal digestions, which illustrated that these of polysaccharide were not hydrolysed under simulated digestion. However, some studies have shown that the Mw of polysaccharides from Ganoderma atrum [19] and Rosa Roxburghii Tratt [14] were significantly decreased under simulated digestion, indicating that some of these polysaccharide could be hydrolysed by simulating upper gastrointestinal fluid. The digestibility of polysaccharides is influenced by their structural characteristics (molecular weight, monosaccharide constituents, and glycosidic linkages) [14]. AABP-2B mainly included (4,6)-β-d-Glc*p*-(1,3,6)-d-Man*p*-(1,4)-β-d-Man*p*-(1) glycosidic linkages [10]. Figure 1 and Figure 2 indicate that the carbohydrate hydrolysis enzymes in the saliva and upper gastrointestinal tract could not break down the glycosidic linkages of AABP-2B. 

### 3.2. Analysis of Inhibition α-Glucosidase Activity after In Vitro Simulation of Digestion

We previously showed the good inhibitory α-glucosidase activity of AABP-2B [10]. Hence, the effects of simulating saliva–gastrointestinal digestion on the α-glucosidase inhibitory effects of AABP-2B were assessed in this study. As shown in Figure 4, the inhibitory effects of AABP-2B and digested AABP-2B on α-glucosidase increased with the increasing concentration. The IC_50_ values of digested AABP-2B and AABP-2B on α-glucosidase inhibition were calculated to be 0.81 ± 0.13 and 0.84 ± 0.19 mg/mL, respectively. These findings demonstrate that saliva–gastrointestinal digestion does not affect (*p* > 0.05) the α-glucosidase-inhibitory activity of AABP-2B, which may be attributed to the largely unchanged structural characteristics of AABP-2B after simulated digestion. α-Glucosidase is an exoenzyme in the intestinal epithelium that can hydrolyse oligosaccharides into glucose. Our results show that digested AABP-2B still possesses good α-glucosidase-inhibitory activity. Hence, the ingestion of AABP-2B may help inhibit glycosidase activity and prevent postprandial hyperglycaemia. 

### 3.3. Change in pH

Indigestible polysaccharides that cannot be digested by gastric or intestinal fluids may be utilised by gut flora [20]. The fermentation process of AABP-2B may be reflected by changes in the pH [21]. As shown in Figure 5, the pH of the AABP-2B and inulin groups decreased sharply during the first 0–12 h, but only slightly during the subsequent 12 h of fermentation. In the AABP-2B group, the pH decreased from 6.89 ± 0.16 (0 h) to 5.12 ± 0.27 (24 h), which was much lower than that of the blank group. These results are similar to those of a previous study on loquat leaf polysaccharides [7]. The decrease in the pH in the AABP-2B and inulin groups may be due to the fact that AABP-2B and inulin act as carbon sources that can be utilized by gut microbes. The bacterial composition may be altered at lower pH levels by encouraging the proliferation of beneficial bacteria and inhibiting the multiplication of pathogenic bacteria [22].

### 3.4. SCFA Content during Fermentation

SCFAs are gut microbe-derived metabolites that may act as important nutrients to promote colon proliferation and maintain homeostasis. As shown in Figure 6F, the SCFAs’ concentration in the AABP-2B group increased profoundly from 1.20 ± 0.31 (0 h) to 32.40 ± 1.14 mmol/L (24 h). Acetic, *n*-butyric, and propionic acids were the dominant metabolic products in the colon (Figure 6A,D). The propionic acid content in the AABP-2B group increased from 0.44 ± 0.13 (0 h) to 16.63 ± 0.43 mmol/L (24 h) (Figure 6B). Moreover, the contents of *n*-butyric and acetic acids considerably increased compared with those in the blank group. Acetate suppresses appetite via the central hypothalamus [23]. Propionic acid is absorbed by the colon, metabolised by the liver to participate in gluconeogenesis, and used as energy. Butyrate, a significant substrate for maintaining the colonic epithelium, not only has anti-inflammatory effects, but also provides energy for colon cells [24,25]. Thus, AABP-2B may be digested by intestinal microorganisms to form beneficial compounds (SCFAs).

### 3.5. In Vitro Fermentation of AABP-2B by the Intestinal Flora

As a carbon source, polysaccharides can provide energy to the intestinal flora and promote its growth, thereby altering the diversity of the intestinal microecology [26]. The sequencing of 16S ribosomal RNA was conducted to investigate the influence of AABP-2B on the intestinal flora during in vitro fermentation. A total of 8364 raw reads were obtained by the sequence optimisation of 30 samples (10 groups, each repeated three times). As illustrated in Figure 7A, the Shannon index curve initially rose dramatically and then reached a plateau, indicating that the sequencing data volume was large enough to reflect the biological information per sample. Figure 7B shows that the rank abundance curves for each sample species are rich and uniform. A high level of alpha diversity indicates that each community’s microbial wealth and uniformity were reflected in the sequencing data. Moreover, the coverage rate for each sample was over 99.87%, suggesting that the data analysis was reasonable (Figure 7C).

Principal component analysis (PCA) plots and hierarchical clustering trees were used to evaluate the differences in microbial composition between samples. After 24 h of fermentation, there was a distinct separation between the four groups (OR, BLK, AABP-2B, and IN), with the first two axes, accounting for 88.19% of the difference between groups (Figure 8A). On PC1 (the first-principal component), compared to the BLK group, the distance between the AABP-2B and IN groups was small, indicating that AABP-2B and IN similarly affected the gut microbiota. Moreover, the cluster analysis results (Figure 8B) are in accordance with the PCA results, illustrating that AABP-2B and IN have similar impacts on the gut microbiota.

At the phylum level (Figure 9), *Bacteroidetes*, *Actinobacteria*, *Proteobacteria*, and *Firmicutes* were the main microbial flora in all samples, which is consistent with a previous report [27]. In the AABP-2B-24 and IN-24 groups, *Firmicutes* and *Bacteroidetes* were more abundant than in the BLK-24 group, whereas Proteobacteria were less abundant. In the colon, *Firmicutes* can ferment non-starch polysaccharides into butyrate, which is beneficial to the host [28]. Moreover, *Bacteroidetes* contain a number of carbohydrate-active enzymes capable of metabolising polysaccharides, which may have the beneficial effect of promoting the growth of other gut flora [28]. *Proteobacteria* are one of the most diverse flora containing not only anaerobic bacteria, but also a certain number of aerobic bacteria [29,30]. After fermentation for 6 h, the abundance of Proteobacteria in the BLK, AABP-2B, and IN groups increased significantly from 2.74% (0 h) to 33.88%, 12.03%, and 7.65%, respectively. However, after 24 h of fermentation, the abundance of Proteobacteria in the BLK, AABP-2B, and IN groups decreased significantly to 10.21%, 3.69%, and 5.71%, respectively. This may be because, although the anaerobic conditions were strictly controlled during the entire fermentation process, there may have been trace amounts of oxygen in the initial fermentation process, which promoted the proliferation of aerobic Proteobacteria [31]. However, with increasing the fermentation time, trace oxygen was depleted and the content of Proteobacteria in each group also significantly decreased, indicating that the entire fermentation experiment maintained good anaerobic conditions. In addition, Proteobacteria comprise many pathogenic bacteria, such as *Shigella*, *Salmonella*, *Escherichia*, and *Campylobacter*, which may result in chronic colitis and inflammation [30]. Compared with the BLK-24 group, the abundance of Proteobacteria in the AABP-2B-24 group was significantly lower, indicating that AABP-2B could inhibit the proliferation of Proteobacteria, which is beneficial to the health of the host.

Compositional changes in the intestinal flora at the genus level are shown in Figure 10A,B. In comparison to the blank group, the abundance of bacteria was very different after fermentation with AABP-2B and inulin, suggesting that fermentation with AABP-2B and inulin had a profound influence on the intestinal flora. Moreover, the relative abundances of *Megasphaera*, *Lactobacillus*, *Megamonas*, *Prevotella*, *Bacteroides*, and *Faecalibacterium* were greatly increased in the AABP-2B group after 24 h of fermentation, while the relative amounts of *Escherichia-Shigella*, *Blautia*, *Dialister*, *Fusobacterium*, and *Romboutsia* were significantly reduced. The abundance of *Bacteroides* significantly increased after AABP-2B fermentation, possibly because of their ability to degrade and utilise carbohydrates, thereby promoting their proliferation [32]. Moreover, *Bacteroides* have a therapeutic effect on obesity-related metabolic and immune system disorders [25,33]. *Faecalibacterium*, such as *Faecalibacterium prausnitzii*, has anti-inflammatory properties and is beneficial to the host [34]. In addition, *Megasphaera* and *Faecalibacterium* may ferment polysaccharides to produce butyrate, which provides energy to colonic epithelial cells and inhibits inflammation [35,36]. After 24 h of fermentation, the AABP-2B group exhibited a higher abundance of *Faecalibacterium* than the blank group. The increase in *Faecalibacterium* was also consistent with the SCFAs results; the butyric acid content in the AABP-2B group was much higher than that in the blank group. In addition, the relative amount of *Prevotella* in the AABP-2B and inulin groups was higher than that in the blank group, indicating that AABP-2B and inulin significantly promote *Prevotella* growth during in vitro fermentation. Previous studies have shown that a high-fibre diet is closely associated with an increase in *Prevotella*, which may degrade polysaccharides and use fermentation products to promote *Prevotella* [37]. Kovatcheva-Datchary et al. observed that an increase in *Prevotella* can promote glycogen storage in some populations and prevent glucose intolerance [33]. Therefore, an in&in *Prevotella* in the intestinal flora may reduce the risk of diabetes.

Moreover, the relative amount of the opportunistic pathogen, *Escherichia-Shigella*, was reduced substantially in the AABP-2B group compared with the blank group. Therefore, AABP-2B may alter the composition of the intestinal flora by inhibiting pathogen growth. In conclusion, *Anemarrhena* polysaccharides (AABP-2B) stimulate the multiplication of probiotics and inhibit the growth of pathogenic bacteria, thus maintaining host health.

## 4. Conclusions

In general, the results show that during digestion, the molecular weight of AABP-2B does not change significantly, and there is no monosaccharide in the digestive juice. These results indicate that AABP-2B is stable under simulated digestion conditions and can enter the large intestine to be utilized by the intestinal flora. Moreover, after salivary–gastrointestinal digestion, AABP-2B still possessed good α-glucosidase inhibitory activity. During AABP-2B fermentation, the pH decreased significantly, whereas total SCFAs, including propionic, acetic, and *n*-butyric acids, significantly increased. Therefore, it is speculated that the changes in SCFAs may be related to the fermentation of AABP-2B by intestinal flora. Additionally, AABP-2B modified the composition of the intestinal flora by increasing the relative amounts of beneficial bacteria, such as *Faecalibacterium*, *Prevotella*, and *Megasphaera*. Furthermore, AABP-2B can alter the gut microbiota composition by inhibiting the growth of pathogens. Thus, AABP-2B may be used as a prebiotic or functional food to improve gut health. Meanwhile, there were certain limitations to this study. As in vitro studies are inherently different from in vivo research, the findings presented herein cannot be directly correlated with biological responses. In the future, we will conduct animal experiments to further investigate the effects of the in vivo digestion and hydrolysis of AAB-2B on the intestinal microbiota and metabolites.

## Figures and Tables

**Figure 1 nutrients-15-01965-f001:**
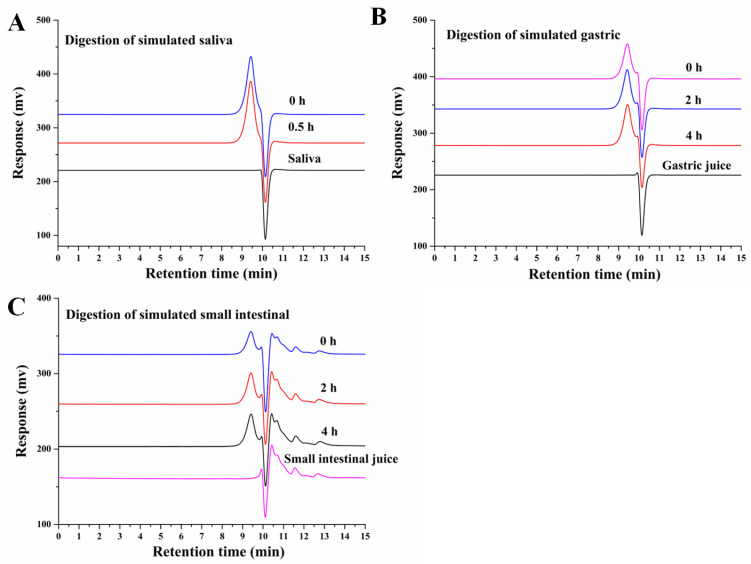
HPGPC chromatograms of polysaccharides. The changes in the molecular weight of AABP-2B during simulated saliva (**A**), and gastric (**B**) and intestinal juice (**C**) digestions.

**Figure 2 nutrients-15-01965-f002:**
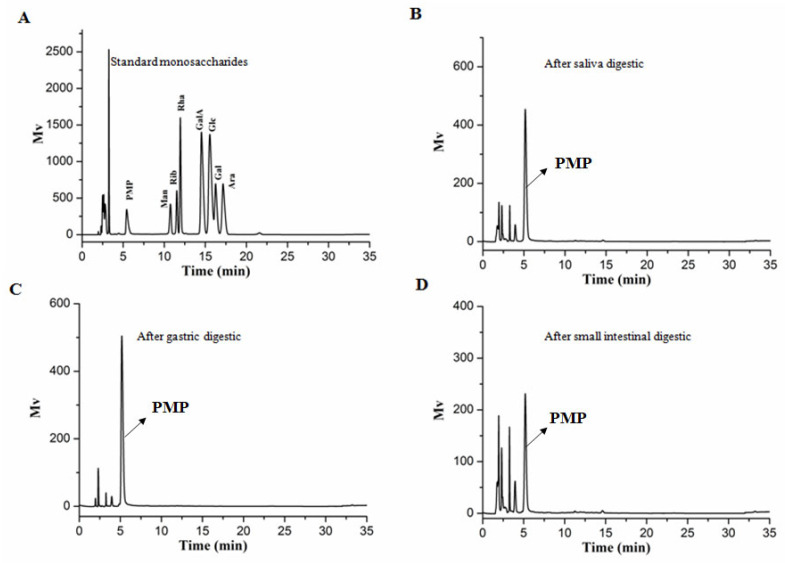
Changes in monosaccharides released from AABP-2B during digestion. Monosaccharide standards (**A**); salivary digestion of AABP-2B for 0.5 h (**B**); gastric digestion of AABP-2B for 4 h (**C**); small-intestinal digestion of AABP-2B for 4 h (**D**). PMP, 1-Phenyl-3-methyl-5-pyrazolone; Man, mannose; Rib, ribose; Rha, rhamnose; GlaA, galacuronic acid; GlcA, glucuronic acid; Glc, glucose; Gal, galactose; Ara, arabinose; Xyl, xylose.

**Figure 3 nutrients-15-01965-f003:**
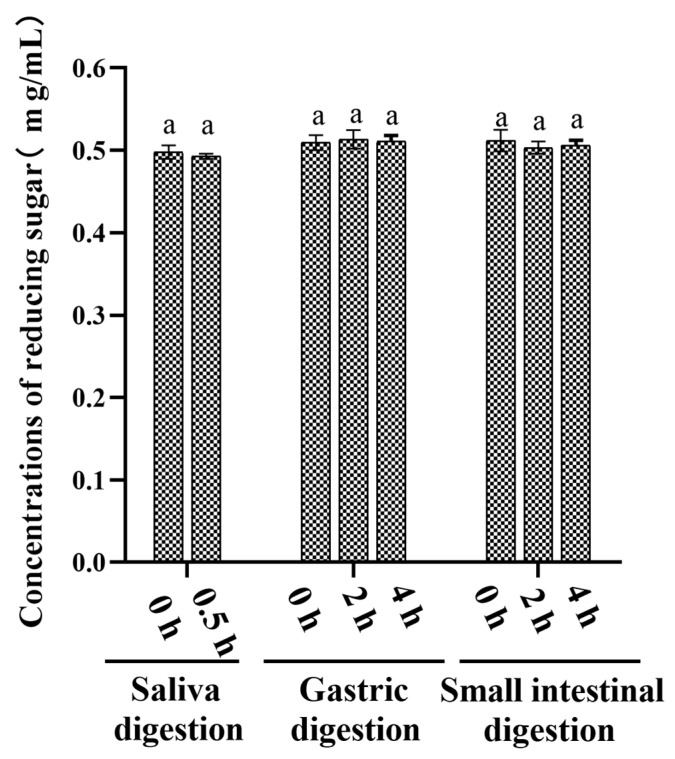
Changes in reducing sugars of AABP-2B during simulation of upper gastrointestinal digestion. Statistically significant differences were observed between different letters (*p* < 0.05).

**Figure 4 nutrients-15-01965-f004:**
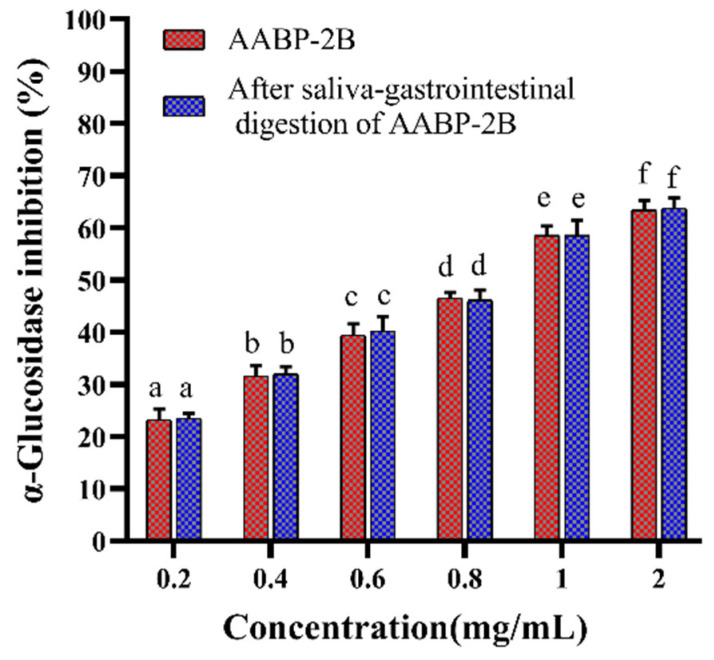
α-Glucosidase-inhibitory activity of AABP-2B during saliva–gastrointestinal digestion. Statistically significant differences were observed between different letters (*p* < 0.05).

**Figure 5 nutrients-15-01965-f005:**
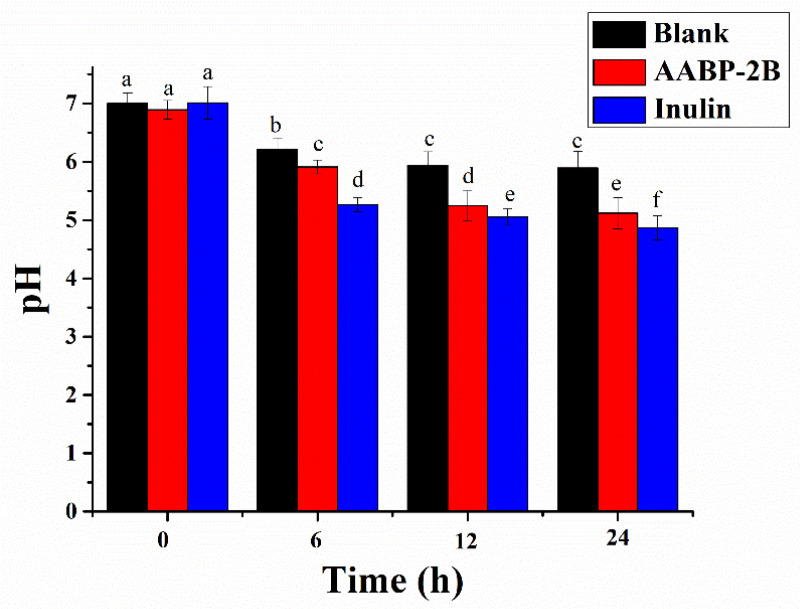
Changes in pH in AABP-2B and inulin groups during fermentation. Statistically significant differences were observed between different letters (*p* < 0.05). Blank, without-carbon-source group; AABP-2B, indigestible *A. asphodeloides* polysaccharide (AABP-2B)-treatment group; IN, inulin-treatment group.

**Figure 6 nutrients-15-01965-f006:**
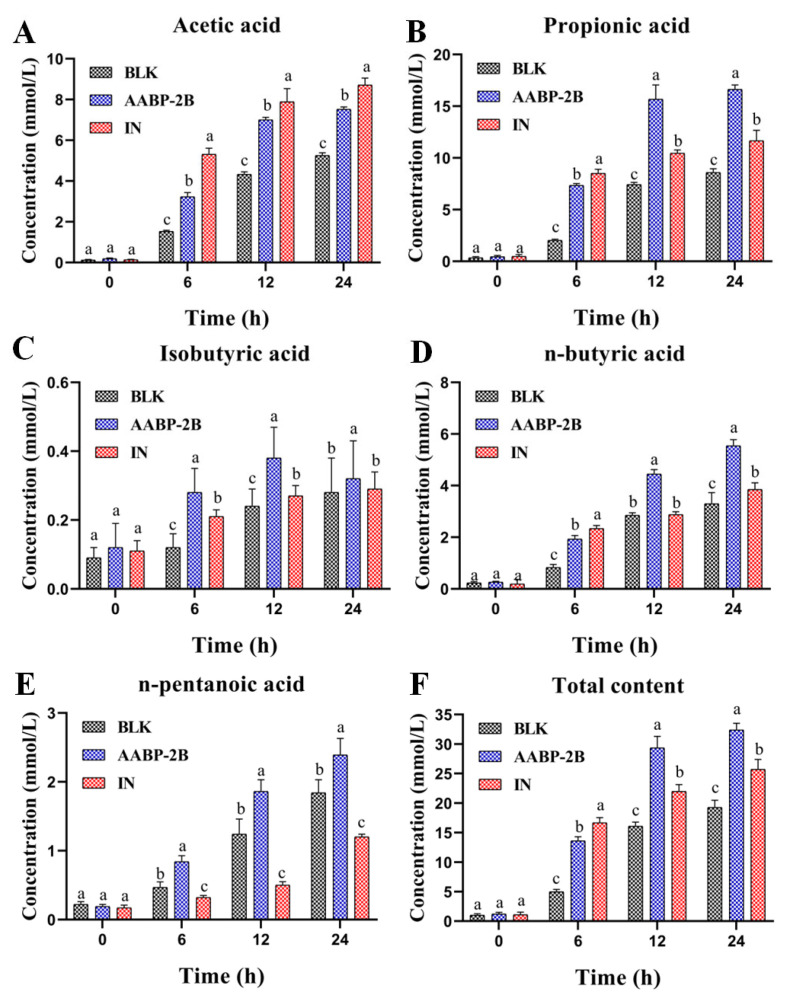
SCFA concentrations of AABP-2B during fermentation: (**A**) acetic acid; (**B**) propionic acid; (**C**) isobutyric acid; (**D**) *n*-butyric acid; (**E**) *n*-pentanoic acid; and (**F**) total content. Statistically significant differences were observed between different letters (*p* < 0.05).

**Figure 7 nutrients-15-01965-f007:**
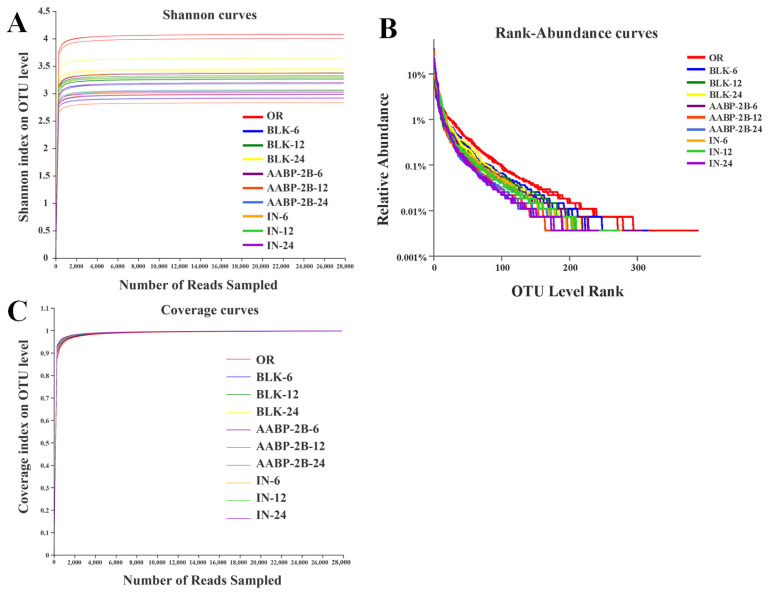
Shannon (**A**), rank abundance (**B**), and coverage curves for each treatment group (**C**).

**Figure 8 nutrients-15-01965-f008:**
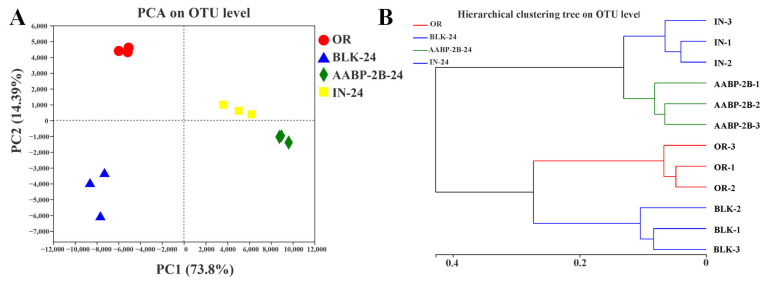
OTU-level PCA of gut microbiota; (**A**) hierarchical clustering tree on OUT level (**B**).

**Figure 9 nutrients-15-01965-f009:**
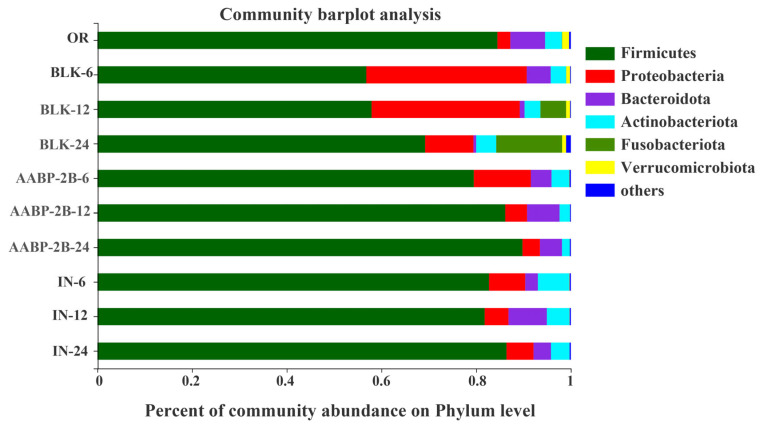
Phylum level distribution of gut microbiota. OR, original faeces group; BLK, without-carbon-source group; AABP-2B, indigestible *A. asphodeloides* polysaccharide (AABP-2B) = treatment group; IN, inulin-treatment group.

**Figure 10 nutrients-15-01965-f010:**
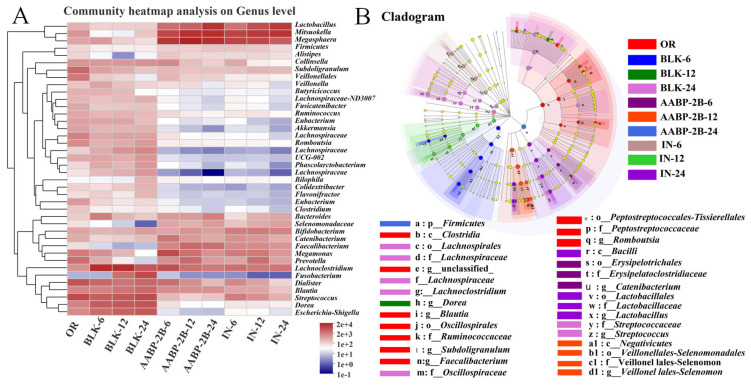
The heatmap analysis of the relative abundance at the genus level (**A**). The linear discriminant analysis effect size (LefSe) analysis in the comparison of gut microbiota among OR, BLK, AABP-2B, and IN control group (**B**). OR, original faeces group; BLK, without-carbon-source group; AABP-2B, indigestible *A. asphodeloides* polysaccharide (AABP-2B)-treatment group; IN, inulin-treatment group.

## Data Availability

Not applicable.

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
