# Peer review of "Effects of Simulated In Vitro Digestion on the Structural Characteristics, Inhibitory Activity on α-Glucosidase, and Fermentation Behaviours of a Polysaccharide from Anemarrhena asphodeloides Bunge"

_nutrients, 2023, doi:10.3390/nu15081965_

Round 1

Reviewer 1 Report

This paper is an interesting investigation of behaviours of polysaharides from Zhimu in in vitro digestion. However, additional clarifications are necessary in the paper.

First, it is not clear whether this research had ethical permission, given that it is working with biological material from people.

Materials and methods must be further clarified - especially in the subsection Statistical analysis, where the software is listed, but if the methods used are not specified (except that the result is expressed as a mean value with the corresponding standard deviations), the repeatability of the study is not ensured.

- the figures are not clearly legible (all of them)

- in figures is the same parameter presented as content (y-axis on figure 3) and in figure 4 it is concentration. Correct is concentration because it is mg/mL and the content would be "mg"

- Fig 10. Title is one heatmap not several -also can not be read what is written on it 

- Figure 5. Is the time frame (24 h) the time when the digestion was observed? Or is this an experiment not related with digestion. If it is related with digestion - add in the figure which part of digestion was observed at what time

- Figure 8. Title - mentioned is the OTU-level, but it is not clarified (either in the Methods or in the discussion, what ...in the same title is mentioned OUT-level, also not clarified.

- if the title highlights "structural changes" - than this must must also be in focus in the discussion and conclusion itself

Sincerely

Author Response

Responses to editor’s and Reviewers’ comments

We appreciate very much for your positive and constructive comments and suggestions on our manuscript entitled “Effects of simulated in vitro digestion on the structural charac-teristics, inhibitory activity on α-glucosidase, and fermentation behaviours of a polysaccharide from Anemarrhena asphodeloides Bunge”. These suggestions are helpful for improvement of this manuscript. We have revised our manuscript according to the comments of reviewers and editors, and the revised parts were highlighted in Red. The main changes in the manuscript and the responses to the comments are in the file named “Detailed Responses to Reviewers”.

We hope this manuscript to be reconsidered by editor and reviewers.

Sincerely

Bing Li

Correspondence: College of Food Sciences and Engineering, South China University of Technology, Wu Shan Road 381, Guangzhou 510640, China.

Detail responses to reviewers are as follows:

1

Comments and Suggestions for Authors

This paper is an interesting investigation of behaviours of polysaharides from Zhimu in in vitro digestion. However, additional clarifications are necessary in the paper.

First, it is not clear whether this research had ethical permission, given that it is working with biological material from people.

Response: Thank the reviewer for the comments. The experimental protocol conforms to the ethical standards set by the committee responsible for human experimentation. Volunteers who provided feces also signed informed consent.

Materials and methods must be further clarified - especially in the subsection Statistical analysis, where the software is listed, but if the methods used are not specified (except that the result is expressed as a mean value with the corresponding standard deviations), the repeatability of the study is not ensured.

Response: Thank the reviewer for the comments. We have added statistical analysis methods in Statistical analysis Section (Line 160-161). In addition, we have added a detailed description of the methods for determining molecular weight, monosaccharide composition, and gut microbiota in revised manuscript (Line 109-121 and Line 148-157).

- the figures are not clearly legible (all of them)- in figures is the same parameter presented as content (y-axis on figure 3) and in figure 4 it is concentration. Correct is concentration because it is mg/mL and the content would be "mg"

Response: Thank the reviewer for the comments. Both the Y-axis on Figure 3 and the X-axis on Figure 4 represent concentration, and we have been reedited the Figure 3 in revised manuscript (Line 196-197). In addition, we have been reedited all figures in the revised manuscript.

- Fig 10. Title is one heatmap not several -also can not be read what is written on it

Response: We thank the reviewer for the kind comment. Since Fig. 10 contains many information on genus-level flora, the writing may not be clear, we have been been reedited the Fig. 10 in revised manuscript (Line 303-306).

- Figure 5. Is the time frame (24 h) the time when the digestion was observed? Or is this an experiment not related with digestion. If it is related with digestion - add in the figure which part of digestion was observed at what time

Response: Thank the reviewer for the comments. Figure 5 shows the pH changes of the polysaccharides of Anemarrhena asphodeloides during in vitro fecal fermentation. We set the fecal fermentation time to 24 hours and detected the pH values at 0, 6, 12, and 24 h during the fecal fermentation process.

- Figure 8. Title - mentioned is the OTU-level, but it is not clarified (either in the Methods or in the discussion, what ...in the same title is mentioned OUT-level, also not clarified.

Response: Thank the reviewer for the comments. The OTU-level refers to the operational taxonomic unit level, which is a classification system used in microbial ecology to group similar genetic sequences based on their similarity. We have added in the detailed description of OTU-level in the Methods (Line 154-157).

- if the title highlights "structural changes" - than this must must also be in focus in the discussion and conclusion itself

Response: Thank the reviewer for the comments. We have added the discussion about structural changes in the revised manuscript (Line 169-171, and Line 180-194).

Reviewer 2 Report

In the present paper the authors studied Anemarrhena asphodeloides Bunge, also called Zhimu, a Chinese herb endowed with a long history in China. The changes of structural properties and the inhibition of α-glucosidase activity of polysaccharide (AABP-2B) were investigated during simulating in vitro salivary-gastrointestinal digestion. The reported results indicated that, under simulated digestion, AABP-2B did not degrade and was able to completely enter the large intestine. At the same time the digestion did not seem to play a relevant effect on the inhibition of α-glucosidas activity mainly ascribable to the basically unchanged structural characteristics of AABP-2B after simulating digestion.  The positive effects of AABP-2B on gut microbial structure was also considered.

Despite the potential interest of the study and the amount of reported data, the study presents many critical issues that must be clarified as follows:

1)               Overall, the paper is written in very bad English. Besides spelling and grammar mistakes, sometimes the sentences don’t contain the main clause, so making difficult to understand their meaning (for example see line 75-77).

2)               The Figures have poor resolution so that it is impossible to distinguish characters (see Fig. 1, overwritten abbreviations make difficult to understand the real meaning of Fig. 2, Fig. 7 and 8 unreadable).

3)               The authors must better specify methods and procedures followed to obtain the results: starting from the Abstract, in the present form the manuscript is too general. The authors always refer to previous furthermore not freely accessible papers.

4)               The structure of AABP-2B must be reported.

5)               Chromatograms of Figure 1 are not described in the text nor in the Caption.

6)               The same for Figure 2. The text and the caption must specify the technique adopted for the analysis as well as describe in detail the peaks and their change.

7)               The same for all the other Figures (text and Captions).

8)               Line 175-76 the authors report the amylase activity as 113 ± 7 units/mL and a range for the experiments much larger than the SD. How can the authors explain this figure, or there was a misleading?

9)               The authors report previous literature reporting dietary not hydrolyzed polysaccharides. This crucial point must be reported in much more detail as concerns the data and much more discussed as concerns the literature. Again, also in this case, the authors merely list the results, and the reader just has to believe them.

10)            Figure 5-6. Statistically significant differences were observed between of different alphabets ( p < 0.05 ). This reviewer cannot understand how the authors decided the alphabets and then the results.

11)            Abstract is too generic and must be rewritten detailing adopted methods and data.

12)            Conclusions section is too short and must be rewritten, even more by considering that the Results and Discussion Sections are treated together so becoming very long.

13)            The limitation of the study (possibly in a separate section) as well as researching prospects must be reported. 

Author Response

Responses to editor’s and Reviewers’ comments

We appreciate very much for your positive and constructive comments and suggestions on our manuscript entitled “Effects of simulated in vitro digestion on the structural charac-teristics, inhibitory activity on α-glucosidase, and fermentation behaviours of a polysaccharide from Anemarrhena asphodeloides Bunge”. These suggestions are helpful for improvement of this manuscript. We have revised our manuscript according to the comments of reviewers and editors, and the revised parts were highlighted in Red. The main changes in the manuscript and the responses to the comments are in the file named “Detailed Responses to Reviewers”.

We hope this manuscript to be reconsidered by editor and reviewers.

Sincerely

Bing Li

Correspondence: College of Food Sciences and Engineering, South China University of Technology, Wu Shan Road 381, Guangzhou 510640, China.

Detail responses to reviewers are as follows:

2

Comments and Suggestions for Authors

In the present paper the authors studied Anemarrhena asphodeloides Bunge, also called Zhimu, a Chinese herb endowed with a long history in China. The changes of structural properties and the inhibition of α-glucosidase activity of polysaccharide (AABP-2B) were investigated during simulating in vitro salivary-gastrointestinal digestion. The reported results indicated that, under simulated digestion, AABP-2B did not degrade and was able to completely enter the large intestine. At the same time the digestion did not seem to play a relevant effect on the inhibition of α-glucosidas activity mainly ascribable to the basically unchanged structural characteristics of AABP-2B after simulating digestion.  The positive effects of AABP-2B on gut microbial structure was also considered.

Despite the potential interest of the study and the amount of reported data, the study presents many critical issues that must be clarified as follows:

1) Overall, the paper is written in very bad English. Besides spelling and grammar mistakes, sometimes the sentences don’t contain the main clause, so making difficult to understand their meaning (for example see line 75-77).

Response: Thank the reviewer for the comments. We have asked a professional editor to correct the grammar and spelling in the revised manuscript. Please see the confirmation that this has been done.

2) The Figures have poor resolution so that it is impossible to distinguish characters (see Fig. 1, overwritten abbreviations make difficult to understand the real meaning of Fig. 2, Fig. 7 and 8 unreadable).

Response: Thanks for your advice. we have been reedited all figures in the revised manuscript.

3) The authors must better specify methods and procedures followed to obtain the results: starting from the Abstract, in the present form the manuscript is too general. The authors always refer to previous furthermore not freely accessible papers.

Response: Thanks for your advice. We have added a detailed description of the methods for determining molecular weight, monosaccharide composition, and gut microbiota in revised manuscript. In addition, we have been reedited Abstract (Line 6-16).

4) The structure of AABP-2B must be reported.

Response: Thanks for your advice. We have added the structure of AABP-2B in the revised manuscript (Line 42-43 and Line 174-175).

5) Chromatograms of Figure 1 are not described in the text nor in the Caption.

Response: Thank the reviewer for the comments. We have added a description of chromatograms of Figure 1 in the revised manuscript (Line 151-152).

6) The same for Figure 2. The text and the caption must specify the technique adopted for the analysis as well as describe in detail the peaks and their change.

Response: Thank the reviewer for the comments. We have added a description of Figure 2 in the text and the caption (Line 159-162).

7) The same for all the other Figures (text and Captions).

Response: Thank the reviewer for the comments. We have been reedited all the Figures (text and Captions).

8) Line 175-76 the authors report the amylase activity as 113±7 units/mL and a range for the experiments much larger than the SD. How can the authors explain this figure, or there was a misleading?

Response: We thank the reviewer for the kind comments. I'm sorry we didn't express clearly and it may have caused misunderstanding. The amylase activity of normal human oral saliva is within the range of 18-208 Dunits/mL, and the amylase activity of the saliva used in this experiment is 113±7 Dunits/mL, which is within the normal range, so the experiment meets the requirements. In addition, we changed the original sentence to " The amylase activity of human saliva used in this study was measured to be 113±7 D units/mL, which is within the normal range of 18-208 D units/mL." in the revised manuscript (Line 149-150).

9) The authors report previous literature reporting dietary not hydrolyzed polysaccharides. This crucial point must be reported in much more detail as concerns the data and much more discussed as concerns the literature. Again, also in this case, the authors merely list the results, and the reader just has to believe them.

Response: We thank the reviewer for the kind comments. We have added more detail abd discussed in the revised manuscript (Line 169-177).

10) Figure 5-6. Statistically significant differences were observed between of different alphabets ( p < 0.05 ). This reviewer cannot understand how the authors decided the alphabets and then the results.

Response: We thank the reviewer for the kind comments. One-way ANOVA and Duncan's test were performed for statistical analysis using the SPSS 22 software (IBM, USA). P < 0.05 was considered a statistical significance. In addition, we have added statistical analysis methods in Statistical analysis Section (Line 143-144).

11) Abstract is too generic and must be rewritten detailing adopted methods and data.

Response: We thank the reviewer for the kind comments. we have been reedited Abstract.

12) Conclusions section is too short and must be rewritten, even more by considering that the Results and Discussion Sections are treated together so becoming very long.

Response: We thank the reviewer for the kind comments. we have been reedited the Conclusions section (Line 295-305).

13) The limitation of the study (possibly in a separate section) as well as researching prospects must be reported.

Response: We thank the reviewer for the kind comments. This study only carried out In vitro simulated digestion and fecal fermentation, so it needs to be further verified by animal experiments, and we will add the researching prospects in the conclusion part.

Round 2

Reviewer 1 Report

I would definitely take the opportunity to praise the authors for making the corrections, because the manuscript is now scientifically clearer.

Please check the English, because for example the first sentence of the summary is very confusing.

Sincerely

Author Response

Comments and Suggestions for Authors

I would definitely take the opportunity to praise the authors for making the corrections, because the manuscript is now scientifically clearer.

Please check the English, because for example the first sentence of the summary is very confusing.

Sincerely

Response: Thank you for your approval and suggestions. The first sentence of the summary have been corrected (Line 317-318).

Reviewer 2 Report

In the light of the changes performed to the text and the improvement of Language and Style, I think the manuscript now suitable for publication in Nutrients journal. 

However in the Caption of Figure 2, the alphabet references to Figure 2B-D must be added.

Author Response

Comments and Suggestions for Authors

In the light of the changes performed to the text and the improvement of Language and Style, I think the manuscript now suitable for publication in Nutrients journal.

However in the Caption of Figure 2, the alphabet references to Figure 2B-D must be added.

Response: Thank you for your approval and suggestions. We have labeled PMP (1-Phenyl-3-methyl-5-pyrazolone) in Figure 2B-D. Additionally, free monosaccharides were not detected during the digestion process, thus cannot be labeled in the Figure 2B-D.